# Incremental Learning of Action Models for Planning

## Jun Hao Alvin Ng and Ronald P. A. Petrick

Edinburgh Centre for Robotics
Heriot-Watt University
Edinburgh  EH14 4AS, Scotland, United Kingdom
{jn68,R.Petrick}@hw.ac.uk

### Abstract

The soundness and optimality of a plan depends on the correctness of the domain model. In real-world applications, specifying complete domain models is difficult as the interactions between the agent and its environment can be quite complex. We propose a framework to learn a PPDDL representation of the model incrementally over multiple planning problems using only experiences from the current planning problem, which suits non-stationary environments. We introduce the novel concept of reliability as an intrinsic motivation for reinforcement learning, and as a means of learning from failure to prevent repeated instances of similar failures. Our motivation is to improve both learning efficiency and goal-directedness. We evaluate our work with experimental results for three planning domains.

## Introduction

Planning requires as input a model which describes the dynamics of a domain. While domain models are normally hand-coded by human experts, complex dynamics typical of real-world applications can be difficult to capture in this way. This is known as the knowledge engineering problem (Cullen and Bryman 1988). One solution is to learn the model from data which is then used to synthesize a plan or policy. In this work, we are interested in applications where the training data has to be acquired by acting or executing an action. However, training data acquired in a planning problem could be insufficient to infer a complete model. While this is mitigated by including **past training data** from previous planning problems, this would be ill-suited for non-stationary domains where distributions of stochastic dynamics shift over time. Furthermore, the computation time increases with the size of the training data.

Following these observations, we present an **incremental learning model** (ILM) which learns action models incrementally over planning problems, under the framework of reinforcement learning. PPDDL, a planning language modelling probabilistic planning problems (Younes and Littman 2004) (see Figure 1), is used for planning, and a rules-based representation (see Figure 2) is used for the learning process. A parser translates between these two representations. Action models that were learned previously are provided to subsequent planning problems and are improved upon acquiring new training data; past training data are not used.

We denote the models provided as **prior action models**. These could also be hand-coded, incomplete models serving as prior knowledge. Using prior knowledge has two advantages: (1) it biases the learning towards the prior action models, and (2) it reduces the amount of exploration required.

While the learning progress cannot be determined without the true action models, we can estimate it empirically based on the results of learning and acting. This empirical estimate, or **reliability**, is used to guide the search in the space of possible models during learning and as an intrinsic motivation in reinforcement learning. When every action is sufficiently reliable, we instead exploit with `Gourmand`, a planner that solves finite-horizon Markov Decision Processes (MDP) problems online (Kolobov and Weld 2016).

Another major contribution of our work is its ability to learn from failure. Actions fail to be executed if their preconditions are not satisfied in the current state. This is common when the model is incorrect. Failed executions can have dire consequences in the real-world or cause irreversible changes such that goal states cannot be reached. `ILM` records failed executions and prevents any further attempts that would lead to similar failure. This reduces the number of failed executions and increases the efficiency of exploration.

The rest of the paper is organized as follows. First, we review related work and then present the necessary background. Next, we provide details of `ILM`. Lastly, we evaluate `ILM` in three planning domains and discuss the significance of various algorithmic features introduced in this paper.

## Related Work

We extend the rules learner from (Pasula, Zettlemoyer, and Kaelbling 2007) that learns a set of relational rules to represent an action which can have probabilistic effects. A relational representation allows generalization of the state space unlike propositional rules which are used in (Oates and Cohen 1996). Our training data consists of state transitions $(s_t, a_t, s_{t+1})$ where $s_t$ is the **pre-state**, $a_t$ is the grounded action, and $s_{t+1}$ is the **post-state**. This requirement is stricter than (Yang, Wu, and Jiang 2005; Zhuo et al. 2010; Cresswell, McCluskey, and West 2013) which learns from sequences of actions with no information on intermediate states. However, these works learn deterministic actions whereas we are interested in probabilistic actions. (Mourão et al. 2012; Martínez et al. 2016) learn probabilistic actions

```
(:action moveCar
  :parameters (?from - location ?to - location)
  :precondition (and (at ?from) (road ?from ?to) (notFlattire))
  :effect (and (at ?to) (not (at ?from))
            (probabilistic 0.25 (not (notFlattire))))
)
```

Figure 1: The PPDDL action model for *moveCar* in the `Tireworld` domain.

```
Action: moveCar(?from ?to)
Name: moveCar
Precondition: at(?from) ∧ road(?from ?to) ∧ notFlattire()
Effect: 0.75 at(?to) ∧ ¬at(?from)
        0.25 at(?to) ∧ ¬at(?from) ∧ ¬notFlattire()
        0 ⟨ noise ⟩
```

Figure 2: The rule for the true action model representing *moveCar*, with two probabilistic effects, in the `Tireworld` domain with arguments *?from* and *?to*.

but do not address the incremental nature of reinforcement learning. (Gil 1994; Wang 1995) learn deterministic action models incrementally while (Rodrigues, Gérard, and Rouveirol 2010) learns probabilistic action models. Our work is most similar to the latter which revises relational rules representing an action whenever contradicting examples are received. They do not store all the examples but rather track how well each rule explains the examples. On the other hand, we address incremental learning over planning problems where only current training data is used. Furthermore, our approach could consider prior knowledge in the form of incomplete action models which can have extraneous predicates unlike (Zhuo, Nguyen, and Kambhampati 2013).

A second area of research that is related to our work is model-based reinforcement learning. `R-MAX` (Brafman and Tennenholtz 2002) is provably sample-efficient, handling the balance between exploration and exploitation implicitly by assigning the maximum reward to unknown states which are set to absorbing states. If the count, defined as the number of times an action is executed in the state, of every applicable action exceeds a threshold, then the state is known. `R-MAX` is impractical for planning problems with large state spaces. Hence, additional assumptions such as factored state spaces (Kearns and Koller 1999), known structures of dynamic Bayesian networks (DBN) (Guestrin, Patrascu, and Schuurmans 2002), or known maximum in-degree of DBNs (Diuk, Li, and Leffler 2009) are often made. Conversely, we only assume that the arguments of actions are known (e.g., we know *moveCar* has *?from* and *?to* as its arguments). We also use an empirical estimate for the learning progress, which we call reliability, as intrinsic motivation. Reliability is also used to quantify prior knowledge which other works on intrinsic motivation do not address (Chentanez, Barto, and Singh 2005; Hester and Stone 2017).

## Background

**PPDDL.** Action models described in PPDDL are defined by their preconditions and effects, typically restricted to con-

junctions of predicates. An example is shown in Figure 1. An action is applicable if its precondition is true in the current state, and executing it changes the state according to its effects which can be deterministic or probabilistic.

**Rules.** For learning action models, we use a rules-based representation as it is well-suited to the incremental nature of reinforcement learning (Rodrigues, Gérard, and Rouveirol 2010). An action is described by a set of rules $R$ where a rule $r \in R$ has three parts: the name of the action, the precondition, and the effect. An example is shown in Figure 2. The key difference between PPDDL and rules representations are the addition of *noise* effect in the latter which serves to avoid modelling a multitude of rare effects which could increase the complexity of synthesizing a plan. When a rare effect occurs, it is often better to replan.

Multiple rules are required to represent disjunctive preconditions or effects. A rule covers a state-action pair $(s, a)$ if it represents $a$ and is applicable in $s$. Every state-action pair in the training data is covered by at most one rule which is called the **unique covering rule**, denoted as $r_{(s,a)}$. A propositional rule is obtained from the grounding of a relational rule by assigning an object or value to every argument in the rule (e.g. grounding *moveCar(?loc1, ?loc2)* to *moveCar(l31, l13)*). Actions are grounded in a similar fashion.

**Markov Decision Processes (MDPs).** MDPs model fully-observable problems with uncertainty. A finite-horizon MDP is a tuple of the form $(\mathcal{S}, \mathcal{A}, T, R, \mathcal{G}, s_0, H)$ where $\mathcal{S}$ is a set of states, $\mathcal{A}$ is the set of actions, $T : \mathcal{S} \times \mathcal{A} \times \mathcal{S} \to [0, 1]$ is the transition function, $R : \mathcal{S} \times \mathcal{A} \to \mathbb{R}$ specifies rewards for performing actions, $\mathcal{G}$ is the set of goal states, $s_0$ is the initial state, and $H$ is the number of decision epochs or planning horizon. The objective is to find a policy which maximizes the sum of expected rewards.

**Reinforcement Learning.** When transition functions in MDPs are not known, model-based reinforcement learning can be used to learn them and perform sequential decision-making. This is the same as learning action models as they can be translated to transition functions (Younes and Littman 2004). Reinforcement learning deals with the balance between exploration and exploitation. Exploration seeks meaningful experiences from which action models are learned while exploitation synthesizes a policy using the models.

## Incremental Learning Model

We propose a new approach to incremental learning across planning problems called `ILM`. `ILM` has two main components: a rules learner and a reinforcement learning framework. We first introduce the concept of reliability which is used in both components followed by the extension made to the rules learner from (Pasula, Zettlemoyer, and Kaelbling 2007). Lastly, we provide details of the framework.

### Reliability of Actions

The reliability of learned action models are empirical estimates of its learning progress. Reliability serves two pur-

poses. We extend the rules learner from (Pasula, Zettle-moyer, and Kaelbling 2007) to consider the prior action model and its reliability to learn new rules. In reinforcement learning, less reliable actions are preferred during exploration. Reliability is defined as:

$$RE(o) = EX(o)\,(\alpha_s\,SU(o) - \alpha_v\,VO(o)) + \gamma^n RE(o_0)$$

where $o$ is an action, $EX$ is the **exposure**, $SU$ is the **success rate**, $VO$ is the **volatility**, $\alpha_s$, $\alpha_v$, and $\gamma$ are scaling parameters, $n$ is the number of updates, and $o_0$ is the prior action model which can be an incomplete action model or an **empty action model** (no predicates in precondition and effect). Reliability is updated whenever $o$ is executed. The initial values of $SU$ and $VO$ are set to zero. The reliability of the prior model is inherited with $\gamma \in (0, 1)$ as the discount factor which reduces its significance given new data.

**Success Rate.** An action with a high success rate indicates that recent executions are successful which is more likely if it has a small error. We define the success rate as:

$$SU(o) = \gamma\,SU(o) + \mathbb{1}(st = success)$$
$$+ 0.5 \times \mathbb{1}(st = partial\ success)$$

where $SU(o) \in \left[0, \frac{1}{1-\gamma}\right)$, $st$ is the execution status, and the indicator function $\mathbb{1}$ equals to 1 if the enclosing condition is true; otherwise, it is 0. The execution status is '*failure*' when the precondition of the action executed is not satisfied. The state is then assumed to be unchanged. The status is '*partial success*' if the post-state is not expected given the learned effects. $SU$ is computed recursively with $\gamma$ as the discount factor which gives less importance to past executions.

**Volatility.** Volatility measures how much a set of rules representing an action changes after learning. A low volatility suggests that learning has converged to the true action model. Volatility is computed recursively, and is defined as:

$$VO(o) = \gamma\,VO(o) + \tilde{d}(R_{prev}, R)$$

where $VO(o) \in \left[0, \frac{1}{1-\gamma}\right)$, $R_{prev}$ ($R$) is the set of rules before (after) learning, and $\tilde{d}(R_{prev}, R)$ is the normalized difference between the two sets of rules. The difference between two rules is defined as:

$$d(r_1, r_2) = d^-(r_1^p, r_2^p) + d^-(r_2^p, r_1^p)$$
$$+ d^-(r_1^e, r_2^e) + d^-(r_2^e, r_1^e)$$

where superscripts $p$ and $e$ refer to the precondition and effect of a rule, respectively, and $d^-(p_1, p_2)$ returns the number of predicates that are in the set of predicates $p_1$ but not in $p_2$. The normalized difference is defined as:

$$\tilde{d}(r_1, r_2) \;\; = \;\; \frac{d(r_1, r_2)}{|r_1| + |r_2|}$$

where the operator $|r|$ refers to the number of predicates in $r$. The difference between two set of rules, $d(R_1, R_2)$, is the sum of differences of pairs of rules $r_1 \in R_1$ and $r_2 \in R_2$ where the rules are paired such that the sum is minimal. Each rule is paired at most once and the number of predicates in unpaired rules are added to the sum.

**Exposure.** Exposure measures the variability (inverse of similarity (Lang, Toussaint, and Kersting 2012)) of the pre-states in the training data, and is defined as:

$$EX(o) = \frac{N_s}{|\mathcal{S}|C_2} \sum_{s_i, s_j \in \mathcal{S}} \frac{d^-(s_i, s_j)}{|s_i|} + \frac{d^-(s_j, s_i)}{|s_j|}$$

where $\mathcal{S}$ is the set of unique pre-states in the state transitions involving $o$, and $N_s$ is the number of state transitions resulting from successful executions. The first term is the ratio of state transitions from successful executions, penalizing those from failed executions which are less informative. Essentially, exposure is the average pairwise difference between pre-states weighted by $N_s$. Since probabilities of effects are inferred using maximum likelihood on the $N_s$ successful state transitions, reliability considers these probabilities implicitly.

Only unique pre-states are used to prevent double-counting. For example, in the `Exploding Blocksworld` domain, the sequence of actions *pickUpFromTable(b1)* and *putDown(b1)* can be executed repeatedly. This also causes $VO$ to decrease and $SU$ to increase which yields a high reliability which does not reflect the learning progress of the actions. Using exposure as a scaling factor prevents such scenarios.

## Learning Rules

The rules learner from (Pasula, Zettlemoyer, and Kaelbling 2007) applies a search operator, selected at random, to a rule. Each search operator modifies the rule differently to yield a set of new rules. An example of a rule is shown in Figure 2. A greedy search uses a score function as heuristics. We introduce a **deviation penalty**, $PEN(R, R_0)$, to the score function such that the search begins from and is bounded around the prior action models, $R_0$, which can be a set of empty rules, or rules of incomplete action models. Hence, the learner refines $R_0$. The score function is defined as:

$$Score(R) = \sum_{(s,a,s') \in \mathcal{T}} log(\hat{P}(s' \,|\, s, a, r_{(s,a)}))$$
$$- \alpha_p \sum_{r \in R} PEN(r) - PEN(R, R_0)$$

where $\hat{P}$ is the probability of the effect in $r_{(s,a)}$ which covers the transition (s, a, s'), $\mathcal{T}$ is the training data, $\alpha_p$ is a parameter, and $PEN(r)$ penalizes complex rules to avoid over-specialization. The deviation penalty increases when $R$ deviates further from $R_0$, and is defined as:

$$PEN(R, R_0) = \frac{RE(o_0)}{EX(o)} \Big[ \alpha_{drop}\,\Delta_{drop}(R, R_0) +$$
$$+ \alpha_{add}\,\Delta_{add}(R, R_0) \Big]$$

where $\alpha_{drop}$ and $\alpha_{add}$ are scaling parameters, and $\Delta_{drop}(R, R_0)$ and $\Delta_{add}(R, R_0)$ are defined as:

$$\Delta_{drop}(R, R_0) \;\; = \;\; \frac{d^-(R_0^p, R^p)}{|R^p| + |R_0^p|} + \frac{d^-(R_0^e, R^e)}{|R^e| + |R_0^e|}$$
$$\Delta_{add}(R, R_0) \;\; = \;\; \frac{d^-(R^p, R_0^p)}{|R^p| + |R_0^p|}$$

where the pairings of rules $r \in R$ and $r_0 \in R_0$ are the same as the pairings in $d(R, R_0)$.

Since past training data is not used, the rules learner may consider a probabilistic effect of $R_0$ as noise if this effect is rarely seen in the current training data. $\Delta_{drop}$ increases when this happens. If the probabilistic effect is not seen at all, it will be dropped in $R$ regardless of how large $PEN(R, R_0)$ is. Such rules will be rejected. The deviation penalty is scaled by the reliability of the prior action model and the inverse of exposure. The intuition is that deviation should be limited if the prior action model is highly reliable, and encouraged if the training data has high variability.

## Planning, Learning, and Acting

We begin this section by explaining the main algorithm for `ILM` (Algorithm 1), followed by the subroutines for reinforcement learning and learning from failure.

The inputs to Algorithm 1 are the prior action models ($R_0$) and their reliability ($RE_0$), initial state ($s_0$), goal state ($g$), and the maximum number of iterations ($N$). $EX_{max} = 0$ and $tabu = \varnothing$ for the first function call and shall be discussed later. The main loop interleaves learning, planning, and acting (lines 5 to 19). Exploration and exploitation is performed at the start of each iteration (line 6). If no action is found, then a dead-end is reached (line 7) and the algorithm terminates. When an action fails to execute, `ILM` learns from this failure by recording the failed instance in $tabu$ (line 11: *relevant_predicates* returns the set of grounded predicates in $s$ that does not contain objects that were not in $a$), otherwise, synthetic state transitions $(s_t, a, s_t)$ are generated (line 13) where $a_T$ is a randomly grounded action such that *check_tabu*$(s_t, a, tabu) \Rightarrow \bot$. Failed executions are exceedingly less as failed instances are added to $tabu$. Reconstructing synthetic failed transitions augment the training data and aids the learning of preconditions.

Learning from training data of low variability (or low exposure) could result in lower correctness of learned rules. To prevent this, we **delay learning** until certain criteria are met (*can_learn* in line 15):

1. If $R_0$ is the set of empty rules, always learn since no information can be lost. However, this risks learning incorrect preconditions or effects that can prevent the agent from reaching the goal state.

2. Otherwise, learn if there is at least one successful transition, at least one failed or synthetic transition, and $EX > \alpha_{EX} EX_{max}$ where $\alpha_{EX} \in [0, 1]$.

If learning is allowed, then new rules are learned (*learn_rules* in line 16) and the values of $RE, EX, VO$, and $SU$ are updated (line 17). Otherwise, only $RE, EX$, and $SU$ are updated. The algorithm terminates after reaching the maximum number of iterations or when the goal is reached. It returns the learned rules, reliability, maximum exposure ($EX_{max}$), and $tabu$. These are used as inputs to the next function call to Algorithm 1.

**Relational Exploration and Exploitation** The balance between exploration and exploitation is implemented in $EE(s, g, R, RE, tabu, \zeta)$. First, we compute the counts

---

**Algorithm 1:** Incremental Learning Model

1  **Function** ILM($R_0, RE_0, s_0, g, N, \zeta, EX_{max}, tabu$):
2      $R \leftarrow R_0$
3      $RE \leftarrow RE_0$
4      $\mathcal{T} \leftarrow \varnothing$
5      **for** $t = 0 : N$ **do**
6          $a_t \leftarrow$ EE($s_t, g, R, RE, tabu, \zeta$)
7          **if** $a_t = \varnothing$ **then break**
8          $s_{t+1}, st \leftarrow$ execute($a_t$)
9          $\mathcal{T}$.append($s_t, a_t, s_{t+1}$)
10         **if** $st = fail$ **then**
11             $tabu$.append(relevant_predicates($s_t, a_t$), $a_t$)
12         **else**
13             $\mathcal{T}$.append(synthetic_transition($tabu, s_{t+1}$))
14         $R_{prev} \leftarrow R$
15         **if** can_learn($R, EX, EX_{max}$) **then**
16             $R \leftarrow$ learn_rules($R_0, \mathcal{T}, RE$)
17         $RE, EX \leftarrow$ update($R, RE_0, \mathcal{T}, st, R_{prev}$)
18         **if** $s' \models g$ **then break**
19     **return** $R, RE, \max(EX, EX_{max}), tabu$

---

for all applicable actions in $s$ using the context-based density formula from (Lang, Toussaint, and Kersting 2012) which performs relational generalizations — the amount of exploration is reduced as states which are unknown under propositional representations could be known under relational representations. The count-action pairs $< c, o >$ are sorted in increasing order of $c = RE(o) \sum_{r \in R} \sum_{(s,a,s') \in \mathcal{T}} \mathbb{1}(r \text{ is applicable in } s)$ in a list, $\mathcal{L}$, where $R$ are rules of $o$. Reliability serves as intrinsic motivation where less reliable actions are explored more.

A state is known if $\forall c_i \in \mathcal{L} \ (c_i \geq \zeta)$, or if the reliability of every action exceeds a constant threshold. The second condition allows exploitation using prior action models when counts are still zero. If the state is known, exploitation is attempted using `Gourmand`, a planner that solves problems modelled in finite-horizon MDP online (Kolobov and Weld 2016). `ILM` can use any planner that accepts planning problems written in PPDDL. Exploitation fails if no plan is found or if the first action of the plan is in $tabu$.

Exploration is attempted if the state is not known or exploitation fails. An action is popped off the top of $\mathcal{L}$ and a list of grounded actions that are applicable in $s$ are enumerated. A grounded action that is not in $tabu$ is selected at random and returned. If no such actions exist, then the next action is popped off until $\mathcal{L}$ is empty, following which random exploration is resorted to where actions are grounded without considering if preconditions are satisfied in $s$. If all grounded actions are in $tabu$, then a dead-end is reached.

**Learning from Failure** Failed executions due to unsatisfied preconditions are recorded in $tabu$. Before an action $a$ is executed in state $s$, Algorithm 2 checks if $(s, a)$ is in $tabu$, returning `False` if so. We describe the algorithm with an example as shown in Figure 3. A state is described by a set of predicates. We extract the set of predicates $f_s \subseteq s$ that does not have an object in its binding that is not in the argu-

**Algorithm 2:** Check if $(s, a)$ is not in $tabu$

```
1  Function check_tabu(s, a, tabu):
2      f_s ← relevant_predicates(s, a)
3      for f_t, a_t ∈ tabu do
4          if name(a) = name(a_t) then
5              f_t ← substitute(f_t, a_t, a)
6              if ∀ p (p ∈ f_s ⇒ p ∈ f_t) then
7                  return false
8      return true
```

$o$: *moveCar(?loc1, ?loc2)*
$a$: *moveCar(l31, l13)*
$s$: *¬hasspare() notFlattire() at(l31)*
   *road(l11 l21) road(l21 l31) road(l12 l11) road(l13 l12)*
   *road(l13 l22) road(l22 l31) road(l22 l21) road(l12 l22)*
   *spareIn(l11) spareIn(l12) spareIn(l21)*
$f_s$: *¬hasspare() notFlattire() at(l31)*
$f_t$: *¬hasspare() notFlattire() at(?loc1) spareIn(?loc2)*
*Perform substitution* $\sigma = \{$*?loc1 → l31, ?loc2 → l13*$\}$ *on* $f_t$
$f_t$: *¬hasspare() notFlattire() at(l31) spareIn(l13)*

Figure 3: An example of checking if $(s, a)$ is in $tabu$.

| Scale | Round | Tireworld | Exploding Blocksworld | Logistics |
|-------|-------|-----------|-----------------------|-----------|
| Small | 1 to 3 | 6 locations | 5 blocks | 2 cities, 4 locations |
| Medium | 4 to 6 | 15 locations | 7 blocks | 2 cities, 6 locations |
| Large | 7 to 10 | 28 locations | 9 blocks | 3 cities, 8 locations |

Table 1: Number of objects in small, medium, and large-scale planning problems for each of the three domains.

| Setting | ILM | ILM-R | ILM-T | R-MAX |
|---------|-----|-------|-------|-------|
| Use Reliability | Yes | No | Yes | No |
| Use $tabu$ | Yes | Yes | No | No |
| Delay Learning | Yes | No | Yes | No |
| Forget Experience | Yes | Yes | Yes | No |

Table 2: Algorithmic configurations for ILM, ILM-R, ILM-T, and R-MAX.

ments of $a$ (line 2). We assume that the arguments of actions are known for this to be possible. $f_s$ is compared to each entry $(f_t, a_t)$ in $tabu$ (lines 3 to 7). The predicates in $f_t$ are grounded with the same substitution as the variables binding of $a$ (line 5). Hence, the check is lifted to relational representations and is applicable even if the objects in the domain change. If $f_s$ does not have at least one predicate that is not in $f_t$, then $a$ is in $tabu$ (line 6). In the example, *moveCar(l31, l13)* is in $tabu$, as are all grounded actions of *moveCar* that do not have *road(?loc1, ?loc2)* in $f_s$. *check_tabu* exploits experiences from failed executions which are otherwise uninformative to the rules learner as it cannot determine the reason for the failure (Walsh et al. 2010). Since every action is checked before execution, $tabu$ will not contain identical entries. This keeps the size of $tabu$ to a minimum which is important as the memory and time complexity is $\mathcal{O}(|tabu|)$.

**Soundness and Completeness** The completeness of Algorithm 2 depends on the failed instances in $tabu$. In the example, if $tabu$ is $\varnothing$, then $a$ is not in $tabu$, and in this case, the algorithm is incomplete. $a$ then fails to execute following which $f_s$ is lifted with $\sigma = \{l31 \rightarrow ?loc1, l13 \rightarrow ?loc2\}$ and inserted with $o$ in $tabu$. Since no erroneous instance is ever added to $tabu$, the algorithm is sound. That is, no action that is found in $tabu$ will succeed in execution.

## Experimental Results

### Experimental Setup

In one trial of experiments, ten planning problems are attempted sequentially in an order of increasing scale (see Table 1). We denote an attempt as one round. Each trial starts with no prior knowledge; the prior action models for round *1* are empty action models. Since the planning problems are probabilistic, 50 independent trials are conducted. The ma-

chine used to run the experiments was a four core Intel(R) i5-6500 with 4 GB of RAM.

We used three planning domains: Tireworld and Exploding Blocksworld domains from the International Probabilistic Planning Competition (Younes et al. 2005), and the Logistics domain. In the Tireworld domain, the car may get a flat tire when moving to another location. If the tire is flat, the car cannot move and a dead-end is reached if no spare tires are available. Tireworld problems of the same scale are identical and are constructed systematically such that there are no unavoidable dead-ends (Little and Thiebaux 2007). In the Exploding Blocksworld domain, a block may detonate when it is put down, destroying the block or table beneath. A destroyed block or table is no longer accessible. Each block can only detonate once. We set the goal states as random configurations of three blocks. All Logistics problems have one truck per city, one airplane, and one parcel. Loading and unloading parcels may fail and the state remains unchanged. The models for all domains are stationary where probabilities of the effects of actions are kept constant in all rounds.

The performance of ILM is evaluated with the correctness of the learned model and the goal-directedness. R-MAX and two variants of ILM are included for comparison. ILM-R does not use reliability; the relational count is not weighted and the deviation penalty in the score function used by the rules learner is zero. In addition, ILM-R does not delay learning (line 15 of Algorithm 1) as this requires $EX_{max}$, a component of reliability. ILM-T does not learn from failure. ILM, ILM-R, and ILM-T do not use past training data while R-MAX does. The algorithmic configurations are summarized in Table 2.

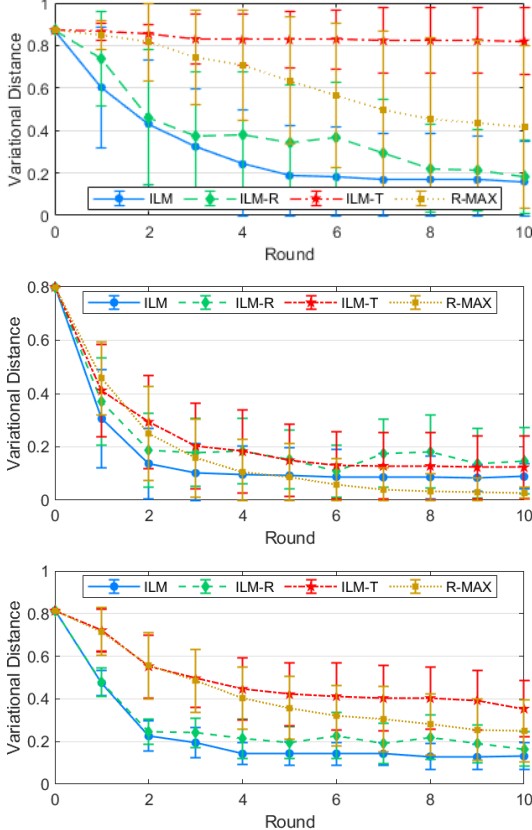

Figure 4: Variational distance at the end of each round for, from top to bottom, `Tireworld`, `Exploding Blocksworld`, and `Logistics` domains. The results are the means and standard deviations of 50 trials.

### Correctness of Learned Models

The correctness of a learned model $\hat{P}$ can be defined as the average **variational distance** between $\hat{P}$ and the true model $P$ (Pasula, Zettlemoyer, and Kaelbling 2007):

$$VD(P, \hat{P}) = \frac{1}{|\mathcal{T}|} \sum_{T_i \in \mathcal{T}} |P(T_i) - \hat{P}(T_i)|$$

where $\mathcal{T}$ is the set of test examples — 500 state transitions per action are generated with the true distribution. Figure 4 show the variational distances for `Tireworld`, `Exploding Blocksword`, and `Logistics` domains. The variational distances at round *0* are of the prior action models, which are empty models for round *1*.

**Tireworld.** `ILM` learns action models incrementally as evident by the decrease in variational distance from rounds *1* to *10*. `ILM-R` performed marginally worse as it learns from training data of low variability which caused the variational distances to increase in rounds *4* and *6*. The utility of learning from failure is illustrated by the significantly larger variational distances for `ILM-T` and `R-MAX`. In both cases, most of the executions led to failure which are less meaningful

experiences for the rules learner. Since the maximum number of iterations is only 15 (*moveCar* alone has 36 possible groundings for the small-scale planning problems), such inefficient exploration performs poorly.

**Exploding Blocksworld.** The lowest variational distances are achieved with `ILM` from rounds *1* to *4* and with `R-MAX` thereafter. The latter learns from a larger training set which is important for this domain which has complex actions *putOnBlock* and *putDown*. These actions have conditional effects which are modelled as separate rules with different preconditions. Preconditions are inferred by comparing prestates in the training data. Most of the predicates in the prestates remain unchanged as an action typically changes a small subset of the state. Hence, more training data is required to learn more complex preconditions. Since the training data used by `R-MAX` are largely from failed experiences, it took four rounds before it outperforms `ILM`.

**Logistics.** `ILM` had the best performance in all rounds. The large variational distances for `ILM-T` is due to the difficulty in learning *driveTruck*. This action has four arguments and there are 432 possible groundings in the small-scale planning problems. This has complications in the goal-directedness which shall be discussed in the next subsection.

### Goal-directedness

The goal-directedness is evaluated by the number of **successful trials** which are trials where the goal state is reached. The goal-directedness for the three domains is shown in Table 3 which underlines the performance of the different algorithmic configurations. It is averaged over rounds with planning problems of the same scale. Round *1* is separated from rounds *2* and *3* to illustrate the advantage of having prior knowledge. The average number of successful trials for rounds *2* and *3* were generally larger than round *1* even though the scales of the planning problems are the same. This is because `ILM` exploits learned models from the previous round whereas round *1* had no such prior knowledge.

**Tireworld.** `ILM-R` outperforms `ILM` in rounds *1* to *3*. This is because the goal state can be reached by executing *moveCar* repeatedly as long as the tire is not flat along the way. `ILM` attempts exploitation more often than `ILM-R`

|  | ILM | | | ILM-R | | | ILM-T | | | R-MAX | | |
| --- | --- | --- | --- | --- | --- | --- | --- | --- | --- | --- | --- | --- |
| **Round** | T | E | L | T | E | L | T | E | L | T | E | L |
| 1 | 18 | 4 | 2 | 22 | 1 | 3 | 3 | 0 | 0 | 6 | 0 | 0 |
| 2 to 3 | 38 | 22 | 10 | 42 | 10 | 5 | 6 | 4 | 0 | 16 | 10 | 3 |
| 4 to 6 | 17 | 27 | 20 | 16 | 12 | 9 | 2 | 9 | 3 | 13 | 8 | 6 |
| 7 to 10 | 6 | 17 | 18 | 4 | 7 | 16 | 1 | 9 | 9 | 2 | 9 | 17 |

Table 3: Average number of successful trials out of 50 trials for `Tireworld` (T), `Exploding Blocksworld` (E), and `Logistics` (L) domains.

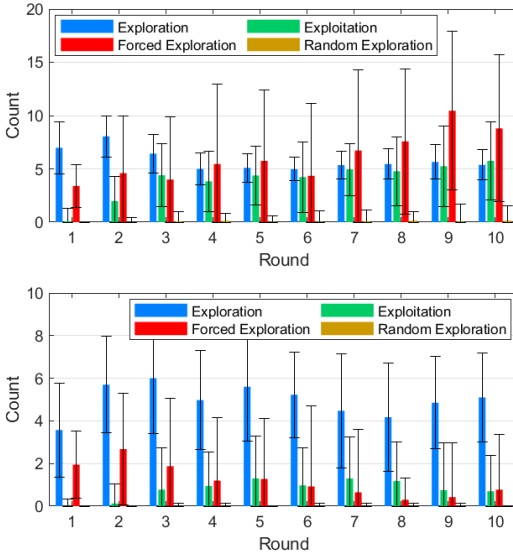

Figure 5: Number of actions from exploration, exploitation, forced exploration, and random exploration that were successfully executed in the `Exploding Blocksworld` domain. The results are the means and standard deviations of 50 trials using `ILM` (top) and `ILM-T` (bottom).

as it weights relational counts with reliability. For small-scale planning problems, exploration or exploitation may not make a significant difference. When the scale increases, the number of steps between the initial state and the goal state increases and the probability of getting a flat tire along the way is higher. A dead-end is reached if the tire is flat and no spare tire is available. In such circumstances, exploitation is required and `ILM` outperforms `ILM-R` in rounds *4* to *10*. `ILM-T` and `R-MAX` did not perform well as actions failed to execute most of the time.

**Exploding Blocksworld.** Dead-ends are often the cause of failing to reach the goal state. A block could detonate with a probability of 0.2 when executing *putDown* or *putOnBlock* which destroys the table or underlying block. These irreversible changes to the state could then lead to dead-ends. Nevertheless, `ILM` has the most number of successful trials in all rounds. `ILM-R` performed much poorer than `ILM` as reaching the goal state with exploration alone is difficult. Even though `R-MAX` has lower variational distances than `ILM` for rounds *5* to *10*, it did not outperform `ILM` as it does not learn from failure. Figure 5 compares the number of actions that were executed successfully in each round using `ILM` and `ILM-T`. The latter had significantly fewer number of successful executions. Figure 5 shows the frequency of exploration decreasing over the rounds while that of exploitation increased. This is expected as the action models are learned incrementally and exploitation should be used in later rounds where the variational distance is lower.

Figure 6 shows the use of $tabu$ in `ILM`. The number of entries added to $tabu$ declined sharply after round *1* because

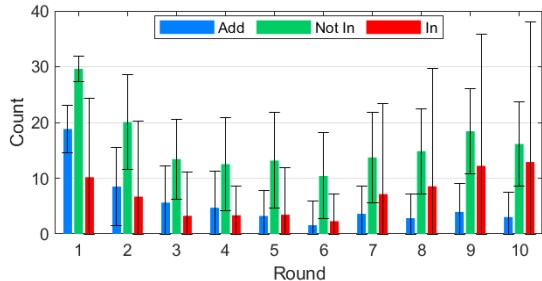

Figure 6: Number of actions added to, found to be in, or found to not be in $tabu$. The results are the means and standard deviations of 50 trials in the `Exploding Blocksworld` domain using `ILM`.

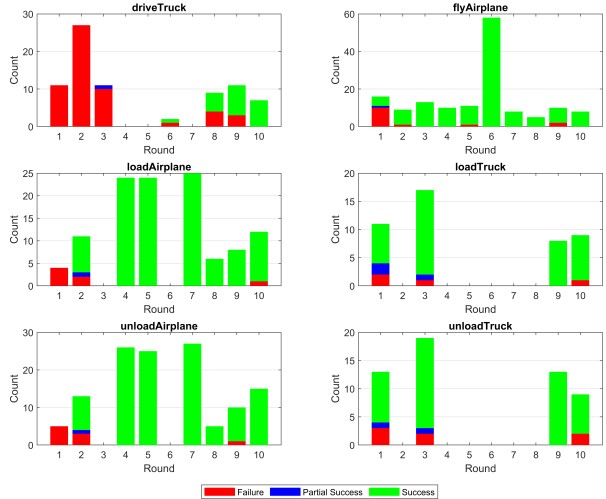

Figure 7: Execution status of actions at each round for a particular trial of `Logistics` domain using `ILM`.

repeating entries are not added. The number of actions found in $tabu$ correspond to the number of failed executions if this check was not done. This number rose for rounds *7* to *10* because the number of grounded actions increases in the large-scale planning problems.

**Logistics.** The number of successful trials increases even when the scale of the planning problem increases. In small-scale planning problems, there were few successful trials because *driveTruck* was not learned yet as mentioned previously. *driveTruck* failed to execute repeatedly till round *3* as only two out of 432 grounded actions would succeed. As a result, a subset of the state space, which could include the goal state, is not reached. If states where the truck is at a location with a parcel are never reached, then *loadTruck* and *unloadTruck* could not be executed. This applies to *loadAirplane* and *unloadAirplane* if a parcel is not at an airport.

For the trial shown in Figure 7, the action model for *driveTruck* was learned from a single partially successful execution in round *3*. The learned rule is shown in Figure 8. Its

*Action: driveTruck(?truck ?from ?to ?city)*
*Precondition: airport(?from) ∧ truck_at(?truck ?from) . . .*
*∧ in_city(?to ?city)*
*Effect: 1.0 ¬truck_at(?truck ?from) ∧ truck_at(?truck ?to)*
*0 ⟨ noise ⟩*

Figure 8: Learned rule for *driveTruck* in the `Logistics` domain. The predicate *airport(?from)* in the precondition is extraneous and causes the action to be erroneously inapplicable in some parts of the state space.

precondition had the extraneous predicate *airport(?from)*. As a result, *driveTruck* is not selected in rounds *4*, *5*, and *7* because this incorrect precondition is not satisfied though the action is in fact applicable. *loadTruck* and *unloadTruck* are not attempted in rounds *4* to *8* because they are in *tabu*. This example illustrates the adverse impact of learning extraneous predicates for preconditions. Although we delay learning when the training data has low variability, this is not done if the action models are empty.

## Conclusions and Future Work

We presented a domain-independent framework, `ILM`, for incremental learning over multiple planning problems of a domain without the use of past training data. We introduced a new measure, reliability, which serves as an empirical estimate of the learning progress and influences the processes of learning and planning. The relational counts are weighted with reliability to reduce the amount of exploration required for reliable action models. We also extended an existing rules learner to consider prior knowledge in the form of incomplete action models. `ILM` learns from failure by checking if an action is in a list of state-action pairs which represents actions that have failed to execute. We evaluated `ILM` on three benchmark domains. Experimental results showed that variational distances of learned action models decreased over each subsequent round. Learning from failure greatly reduces the number of failed executions leading to improved correctness and goal-directedness.

For complex domains, more training data is required to learn action models. Using past training data would not work well for non-stationary domains and also increases the computation time for learning. The first issue could be resolved by learning distributions from the current training data only. The second issue could be resolved by maintaining a fixed size of training data by replacing older experiences while maximizing the exposure, or variability, of the training data. These will be explored in the future.

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
