# OpenReview forum: "Incremental Learning of Action Models for Planning"
_icaps-conference.org/ICAPS/2019/Workshop/KEPS — KEPS 2019_

### Official Review · AnonReviewer2 · 2019-05-08
**An interesting paper on a framework  for incremental learning over planning problems in PPDDL**

**Rating:** 4
**Confidence:** 2

**Review:**

This paper presents a domain-independent framework, ILM (incremental learning model), for incremental learning over multiple planning problems of a domain in PPDDL (without the use of past training data).

ILM learns from failure by checking if an action is in a list of state-action pairs which represents actions that have failed to execute.

ILM is evaluated on three benchmark domains, showing that learning from failure greatly reduces the number of failed executions leading to improved correctness and "goal-directedness".

The paper is moslty well written, ineteresting and fitting the workshop topic.

Some suggestions/questions:

- Model in figure 1. I think it is a bit misleading, especially because of the inverse modeling of flat tire predicate. This probably comes from the IPC formulation, but would be way more intuitive if modeled with a Flattire() predicate instead of notFlattire(), having 0.25 probability that as an effect it gets true.

- p3, Reliability of actions. Please gives 2 words of explanation for exposure and volatility when the first formula is presented. Notwithstanding they are more extensively defined later, it is needed to grasp the meaning of the formula (elaborate also more on the whole formula would also help).

- Why "In reinforcement learning, less reliable actions are preferred during exploration"?

---

### Official Review · AnonReviewer1 · 2019-05-11
**A framework for incremental PPDDL action model learning with thorough evaluation**

**Rating:** 4
**Confidence:** 2

**Review:**

The paper presents an approach for incrementally learning a probabilistic domain model in PPDDL. The motivation is that specifying complete domain models is challenging in one shot and that this problem is compounded in non-stationary environments. The paper introduces a novel framework, which is supported by a solid evaluation. Whereas the main issue with the current approach is learning overly conservative preconditions, it seems that this should be addressable within the current framework.

The approach to tackling the non-stationary environments is to only use the current example for generating training data for learning. Previous learning is communicated through a previous domain model, a reliability measure (to indicate the completeness of the previous model) and a list of state action pairs that led to failed action execution. This framework leads to a clean separation between past examples and derived knowledge and the current learning example. However, it does seem quite a specific scenario and I didn't find a convincing argument for why no previous data is retained. E.g., what happens if the framework is exposed to a small problem?

The factors involved in the reliability score seem reasonable, but little discussion is provided. This section as a whole would be improved with more motivation and better building of intuition. There are lots of parameters here and how any of these are set or how they should be set in a new environment is not clearly discussed.

The results indicate that the approach provides an effective learning framework. The analysis of the contribution of different factors is particularly informative. Some suggestion of how important using Gourmand for exploitation would be interesting, particularly in relation to Table 3. But overall, the current analysis of stationary environments is thorough and provides a solid base. It will be interesting to see how it acts in non-stationary environments and how the collection of parameters impacts on this.